# Metabolic Dysfunction in Spinal Muscular Atrophy

**DOI:** 10.3390/ijms22115913

**Published:** 2021-05-31

**Authors:** Marc-Olivier Deguise, Lucia Chehade, Rashmi Kothary

**Affiliations:** 1Regenerative Medicine Program, Ottawa Hospital Research Institute, Ottawa, ON K1H 8L6, Canada; mdegu051@uottawa.ca (M.-O.D.); lcheh087@uottawa.ca (L.C.); 2Department of Pediatrics, Children’s Hospital of Eastern Ontario, Ottawa, ON K1H 8L1, Canada; 3Centre for Neuromuscular Disease, University of Ottawa, Ottawa, ON K1Y 4E9, Canada; 4Department of Cellular and Molecular Medicine, University of Ottawa, Ottawa, ON K1H 8M5, Canada; 5Department of Biochemistry, Microbiology, and Immunology, Faculty of Medicine, University of Ottawa, Ottawa, ON K1H 8M5, Canada; 6Department of Medicine, University of Ottawa, Ottawa, ON K1H 8M5, Canada

**Keywords:** fat, glucose, metabolism, nutrition, body composition, amino acids, gastrointestinal

## Abstract

Spinal muscular atrophy (SMA) is an autosomal recessive genetic disorder leading to paralysis, muscle atrophy, and death. Significant advances in antisense oligonucleotide treatment and gene therapy have made it possible for SMA patients to benefit from improvements in many aspects of the once devastating natural history of the disease. How the depletion of survival motor neuron (SMN) protein, the product of the gene implicated in the disease, leads to the consequent pathogenic changes remains unresolved. Over the past few years, evidence toward a potential contribution of gastrointestinal, metabolic, and endocrine defects to disease phenotype has surfaced. These findings ranged from disrupted body composition, gastrointestinal tract, fatty acid, glucose, amino acid, and hormonal regulation. Together, these changes could have a meaningful clinical impact on disease traits. However, it is currently unclear whether these findings are secondary to widespread denervation or unique to the SMA phenotype. This review provides an in-depth account of metabolism-related research available to date, with a discussion of unique features compared to other motor neuron and related disorders.

## 1. Background

Spinal muscular atrophy (SMA) is a degenerative neuromuscular disease of the lower motor neurons in the anterior horn of the spinal cord. It is primarily characterized by progressive weakness and wasting in the limb, respiratory, and bulbar muscles. Clinical hallmarks include fasciculations, altered reflexes, joint contractures, and difficulty breathing and feeding [1]. The estimated incidence of SMA is 1 in 11,000 live births with a reported carrier frequency of 1 in 40, making it one of the most common inherited neurodegenerative diseases [2,3]. In SMA, a homozygous deletion or point mutation in the *Survival motor neuron 1* (*SMN1*) gene leads to loss of SMN production from this gene [4]. However, a paralogous copy of *SMN1*, named *SMN2*, is present on the same chromosome [5]. *SMN2* undergoes alternative splicing of exon 7 in 85% of the transcripts primarily due to a one nucleotide difference in this exon [5]. This results in the production of only 5–10% of full-length functional SMN protein from *SMN2* [5]. The copy number of *SMN2* is variable and inversely proportional to the age of onset, as well as the severity of SMA disease [6]. Thus, the phenotypic spectrum in SMA ranges from prenatal symptom onset to mild proximal weakness in ambulatory adults and eventual loss of ambulation. Based on onset and severity of disease, patients are classically classified into types (I to IV) and physical milestone reached (non-sitters, sitters, and walkers).

The pathogenesis of SMA remains poorly understood. The ubiquitously expressed SMN protein plays a significant housekeeping role in pre-mRNA splicing and many other cellular processes [7,8,9,10]. The α-motor neurons appear preferentially affected by SMN protein deficiency. However, accumulating evidence has challenged this traditional paradigm. SMN depletion is also detrimental to other tissues and organs, affecting whole-body physiology [11,12,13,14,15,16,17,18,19,20,21,22,23,24,25,26,27,28,29,30,31,32,33,34,35]. Most recently, metabolic and endocrine organs have also been reported to show altered function in SMA. Early research focused on lipid disturbances in different forms of spinal muscular atrophies [36,37,38,39]. Several new studies have highlighted numerous metabolic alterations in SMA patients and animal models of the disease [30,31,40,41]. However, it should be noted that a clear mechanism linking SMN depletion and the various metabolic abnormalities is currently lacking. Nevertheless, such defects could have serious clinical implications for SMA patients.

To note, several mouse models have been generated to study SMA due to the exclusivity of the *SMN2* gene to humans. Briefly, the “severe model” (*Smn^−/−^;SMN2)* and the “delta 7 model” (*Smn∆7* or *Smn^−/−^;SMN2^+/+^;Smn∆7^+/+^**)* were generated via deletion of the mouse *Smn* gene and introduction of the human *SMN2* gene and have a mean survival of 5 and 14 days, respectively [42,43]. The “Taiwanese model” *(Smn^∆7/∆7^;SMN2^+/+^*) has the mouse *Smn* gene with a knock-out of exon 7 (known as delta 7) with the human *SMN2* transgene in varying copy number [44]. The “2B/- model” (*Smn^2B/^*^−^) contains an engineered three-nucleotide substitution mutation within the exon splice enhancer in exon 7 of the mouse *Smn* gene and has a median survival of 28 days [45].

This comprehensive review aims to highlight the current evidence concerning metabolic dysfunction in SMA. We take a top-down approach, starting our discussion with baseline body composition, quality and quantity of intake, potential issues in the gastrointestinal (GI) tract, and discuss the handling of macronutrients at the molecular level. Furthermore, we offer a comparison to other relevant neuromuscular diseases to underscore the uniqueness of the metabolic phenotype observed in SMA. Knowledge regarding the potential clinical implications of altered metabolism in SMA is also presented. Lastly, we put forward some speculative mechanisms to explain the SMA metabolic presentation and provide future avenues of research in this area.

## 2. Baseline Anthropometric Data and Energy Expenditure of SMA Patients

Body composition and nutritional status are significantly altered in SMA patients. Across all types, both under- and overnutrition are reported [46,47,48,49]. Thus, prescribing tailored energy requirements for SMA patients is challenging due to various possible nutritional phenotypes. Despite meeting adequate caloric intake recommendations, some patients will experience excessive weight gain while others are at risk of undernutrition [48,50,51,52]. Across SMA types II and III, adjusted for age and sex, 56% of patients had body mass index (BMI) below the 50th percentile and 20% had a BMI above the 85th percentile [46]. Meanwhile, insufficient weight gain is reported in SMA type I [53], while malnutrition, especially in younger patients, is seen [52]. In SMA type II patients, weight gain is thought to result from a reduced level of activity and lower resting energy expenditure [54,55]. In contrast, SMA type I patients have the increased work of breathing along with feeding and swallowing problems, contributing to reduced caloric intake and increased energy expenditure [50,51]. From a preclinical perspective, sedentary *Taiwanese (Smn^∆7/∆7^;SMN2^+/+^*) mice, a model of severe SMA, similarly have higher energy expenditure than control animals at 12 months of age [56].

Assessing body composition in the pediatric population is challenging, and the SMA population adds its own intricacies. Some patients with acceptable fat mass (FM) may plot as underweight since the BMI underestimates their body fat [50,51,57]. Relative to age and sex, these patients have a higher fat mass index (FMI) and lower lean tissue, which puts them at a higher risk of overweight status when compared to healthy children [46,54,57,58,59]. Across SMA groups, fat-free mass (FFM) and lean body mass (LBM) distribution are also different. Children with SMA type I have lower FFM and LBM in the trunk and arms specifically, compared to children with SMA type II [54]. Total body water in SMA type I and SMA type II patients is significantly lower than the reference values for sex and age [54]. FFM is a multi-component measure that includes water, protein, minerals, and glycogen [60]. Thus, the lower FFM reported in these studies could result from differing FFM components with healthy peers.

Lack of accuracy surrounding assessment of body composition and metabolic parameters exists [46,54,55]. Therefore, difficulty with the interpretation of these data may lead to conclusions of overfeeding or underfeeding. Thus, patients who are reported as being underfed may in fact have not been, leading to overfeeding and consequences such as increased FM and worsening conditions [54]. Considering clinical features (ventilatory status and nusinersen treatment), a recent study put forward new energy equations to predict energy expenditure for SMA type I patients in the hope of circumventing these issues [61]. However, the authors still recommend indirect calorimetry to ensure better individualized support [61]. A more detailed analysis of some of these studies has recently been published [62]. Additional work will be required to ensure that we harness these objective measures to support and optimize nutritional status for the evolving SMA population.

## 3. Feeding Difficulties and Nutritional Intake Issues

Feeding and swallowing problems have been reported across all types of SMA due to bulbar dysfunction [51]. These issues are common in sitters (SMA type II) and non-sitters (SMA type I), with a 36% prevalence of at least one feeding-related difficulty [63]. Commonly reported problems are jaw issues, fatigue associated with mastication, and choking [64,65]. Opening of the mouth and difficulty with chewing are among other limitations reported by SMA patients [53,65]. Many SMA patients have modified feeding preparation and gastrostomy tubes to aid with feeding due to a risk of aspiration, pulmonary infections, and recurrent choking episodes. No doubt, these problems pose challenges to attaining adequate nutritional intake. Along with prolonged mealtimes and these feeding modifications, patients have reduced food intake, leading to growth failure [53].

Dietary and caloric intake in SMA patients may be lower than what is recommended for healthy peers due to decreased demands [47,66]. Thus, the inability to achieve appropriate micronutrient intake puts SMA patients at risk for crucial micronutrient deficiencies [66]. Vitamins A, D, E and K, folate, calcium, and magnesium are frequently below the recommended ranges [47,66,67]. Deficiencies in vitamin D, a micronutrient relevant to bone health, are commonly reported to be lower at baseline in SMA patients [48,66]. Vitamin D levels were in the lower range of normal, and 50% of the children had low bone density [67]. Combined with these deficiencies, reduction in muscle mass, activity, and ambulation in SMA patients influences bone mass and architecture [68], which also puts SMA patients at higher risk of osteopenia, osteoporosis, and fractures [47,50,66]. Children with SMA type II and III can have sustained fractures and asymptomatic fractures [67,69]. While micronutrient intake is sub-optimal in the SMA population, the metabolism of these substrates also appears to be abnormal. Severe *Smn^−/−^;SMN2* mice have a significantly increased osteoclast formation and bone resorption rate compared to wild-type mice [70]. Alternatively, abnormalities in calcium pathways have been reported in SMA (muscle [71,72], astrocytes [29,73,74], motor neurons [75,76,77], kidney [32]) and abnormal calcium levels have been inconsistently identified in SMA patients [32,47,67]. The specific mechanism and details of these studies are outside the scope of this review. Sufficient, yet careful, vitamin D and calcium supplementation are necessary for maintaining bone density and health in this population, especially considering potential intrinsic abnormalities to those pathways [50,51,78].

## 4. Gastrointestinal Dysfunction

SMA patients can also have gastroesophageal motility issues, which may present as constipation, bloating, and gastroesophageal reflux disease. It is important to note that with the progression of the disease, many SMA patients are dependent on the use of ventilators such as the Bilevel Positive Airway Pressure (BiPAP), which can lead to gastric insufflation. This can worsen gastric emptying [79,80]. Investigations have shown that SMN deficiency in mice is linked to disruptions in the enteric nervous system leading to esophageal and intestinal motility issues [81]. Nestin-F7 (*SMN2^+/+^*; *SMN*Δ7*^+/+^;Smn^F7/−^;nestin-cre*) and *Smn∆7* mice had drier and infrequent fecal pellets, indicating signs of constipation [81]. They also had issues with slower gastric emptying, lengthened intestinal transit, and abnormal colonic motility, without alterations in the morphology of the GI tract [81]. On the other hand, histopathological defects in the GI tract were observed in the *Taiwanese (Smn^∆7/∆7^;SMN2^+/+^*) mice [82,83,84]. They had disproportionately long intestines, alteration in the size, shape and distribution of intestinal crypts, as well as diffuse edema and macrophage infiltration in the lamina propria of the small intestine [82]. Sintusek et al. (2016) also observed a 65–75% reduction in blood vessel density in the intestine, hinting that vascular defects secondary to SMN deficiency may be responsible for such pathological changes [82]. Wan et al. (2018) hypothesized that the damage to the intestinal barrier could lead to translocation of gut bacteria into the blood of SMA mice, explaining the systemic inflammation observed in their study [83]. Thus, it appears that SMN may play an important role in the integrity of the GI tract in mouse models of SMA. While these findings may partly explain slow gastric motility in SMA patients, they are likely to also contribute to disruption of optimal nutrient absorption. Such findings are yet to be confirmed in human pathology. Standard of care recommendations for SMA patients suggest the use of a high-fiber diet and probiotic to aid in the management of constipation and associated GI dysmotility symptoms [51]. With the emerging evidence linking gut bacteria to neurodegenerative diseases and movement disorders, the use of probiotics in modulation of gut microbiota may potentially improve this component of the SMA phenotype [85,86,87,88].

## 5. Intermediary Metabolism in SMA

Thus far, it seems clear that SMA patients, due to their condition, are at risk of altered body composition, sub-optimal caloric intake and nutrition. Moreover, recent evidence highlights that the mucosal barrier of the GI tract is also affected, which may lead to deficiencies in the absorption of micro and macronutrients. Altogether, these foster a situation of altered homeostasis of micro and macronutrients. Interestingly, disruption in macronutrient (fat, glucose, protein/amino acid) handling is also evident and most research to date has focused on this aspect. A comparison to other diseases with denervation is provided below (Figure 1).

### 5.1. Fat Metabolism

Metabolic abnormalities were identified in SMA before the genetic basis of the disease was established. The first suggestive feature appeared in a small case series as dicarboxylic aciduria in some, but not all, SMA patients who were undergoing a fasting period [38,39,89]. Additional significant findings included reduced activity of various enzymes (short-chain l-3-hydroxyacyl-CoA dehydrogenase, long-chain l-3-hydroxyacyl-CoA dehydrogenase, acetoacetyl-CoA thiolase, and 3-ketoacyl-CoA thiolase), increased ratio of dodecanoic to tetradecanoic acid, normal production of ketone, and variable acylcarnitine profile [38,39,89]. An elevation in esterified:free carnitine ratio was also observed [39], linking earlier findings of reduced SMA muscle carnitine levels and carnitine palmityl transferase activity [90]. The authors suggested that this may be independent of denervation, as denervated controls were not similarly affected in some measures [38,89]. Similarly, the pattern of abnormalities in serum fatty acid suggested a unique defect compared to mitochondrial defects known [38]. The findings were generally more prominent in severe SMA patients [38,39]. In these three reports, a few SMA patients had liver steatosis, a feature also recently reported [38,39,89,91]. In another study, 10 of 14 patients had serum carnitine deficiency, while six out of 10 had a similar decrease in carnitine in the muscle, in line with previous reports [39,92]. A thorough analysis of early reports was recently reviewed [62]. The underlying pathogenic mechanism causing these defects remains unclear.

Recently, there was a study aimed at renewing our understanding of alterations in fat metabolism. SMA patients displayed one measure of dyslipidemia about twice more commonly than published pediatric data sets [31]. Similarly, nearly 13% of the cohort of 72 SMA patients had three or more measures of dyslipidemia, where no baseline epidemiological data are currently available in the average pediatric population [31]. It should be noted that abnormal lipid levels were identified upon one blood draw. The chronicity of these abnormalities for an individual is lacking. These findings were distributed evenly across all SMA types, without clear dependence on severity [31]. A recent study in older SMA type II and type III patients similarly identified that about 30% had at least one abnormal lipid level reading [93]. These results are reminiscent of previous studies as well. Eleven out of 12 patients from an older cohort of clinically diagnosed SMA patients had lipid abnormalities [36]. However, the mean age of the patients was 41.5 years [36], compared to 3.8 years in the recent study [31]. There is another study from 1970 of older patients with symptomatology similar to SMA and concomitant hyperlipoproteinemia [37]. These older data sets should be interpreted with caution, given the lack of genetic diagnosis at the time of the studies, making it difficult to adequately define and associate the phenotype to the genetic basis in those particular patients.

Of interest, three out of eight autopsies of SMA patients (most below 14 months of age) showed microvesicular steatosis [31], adding to the previous count of five patients identified with fatty liver [39,89,91,94,95]. The prevalence of liver steatosis in 2 to 4-year-old children that are not affected by SMA is only 0.7% [96]. This is a marked difference, especially when considering that the population studied by Deguise et al. (2019) was even younger [31]. More recently, 11% of SMA type II and III cohorts showed evidence of steatosis through non-invasive imaging such as ultrasound [93]. Furthermore, it is interesting that liver fat content was abnormal in four different mouse models of varying severity [31]. More particularly, severe mouse models had less fat in their liver, while models of milder severity had an accumulation of triglycerides (TGs) in their liver [31]. Of these models, the *Smn^2B/−^* mice invariably developed non-alcoholic fatty liver disease (NAFLD) and dyslipidemia while on normal chow [30,31]. Interestingly, only minor changes were identified in fat classes and chain length in SMA disease-relevant tissues such as the spinal cord and the muscle in pre-clinical models, pointing to the possibility that this may be restricted to the liver or the metabolic organs [31]. However, a recent study using proteomic analysis found metabolism and gene expression as one of the clusters in Gene Ontology (GO) analysis in the CSF of SMA patients [97]. It also identified Apolipoprotein A1 and Apolipoprotein E to be misregulated and that nusinersen may provide a positive effect [97]. Unfortunately, there are no CSF studies in animal models to allow for similar comparison. Nevertheless, *Smn^2B/−^* muscle, another disease-relevant tissue, also has increased fat droplets upon ultrastructural analysis and widespread changes in a fatty acid specific gene expression PCR array [31,98]. Note that it is unclear whether this is a secondary consequence of denervation and atrophy of the muscle.

The cause of fatty liver and dyslipidemia in SMA remains unknown. Despite showing wide proteomic changes in mitochondrial health and function, mitochondrial dysfunction is unlikely, as the isolated hepatic mitochondria from *Smn^2B/−^* mice function better than control [30]. This was suggested to be compensatory to the high TG content in the *Smn^2B/−^* livers in order to enhance clearance and restore homeostasis. Interestingly, a recent study showed that Taiwanese *(Smn^∆7/∆7^;SMN2^+/+^*) mice have a global shift in their respiratory exchange ratio, hence relying more heavily on β-oxidation [56]. The *Smn^2B/−^* hepatic mitochondria produce more reactive oxygen species, leading to liver damage and functional deficit (protein production, hemostasis, complement, iron metabolism, insulin-like growth factor 1 (IGF1) pathway) [30]. Noteworthy, microsomal oxidation, an alternative oxidative pathway turned on upon β-oxidation overload, produces dicarboxylic acids [99]. Its main enzyme, cytochrome P450 4A (Cyp4A), was induced in the *Smn^2B/−^* liver. The confirmation of microsomal induction in *Smn^2B/−^* liver may offer an appropriate explanation for the observation of dicarboxylic aciduria in SMA patients [38,39,89]. Tein et al. (1995) speculated on a potential mechanism for the excess dicarboxylic acid production, which has thus far been in line with the current evidence available in most recent studies [30,39]. Liver intrinsic defects may predispose SMA preclinical models and patients to NAFLD. Interestingly, the findings described above lead to the potential new role of SMN-depleted mice as models for NAFLD [30]. Features of dyslipidemia, fatty liver, insulin resistance, and functional deficits are all well described entities in NAFLD pathogenesis [30]. In a recent study, the liver was the organ with the most disrupted transcriptome in SMA [100]. Nevertheless, it is most likely that pressures from other metabolic organs such as the pancreas, skeletal muscle, adipose tissue, and GI tract contribute to the pathogenesis of NAFLD in SMA (see Section 7 on “Metabolic crosstalk and proposed mechanisms”). Interestingly, a newly described very mild model of SMA (similar to SMA type IV) called the *Smn^2B/−^;SMN2^+/−^* mouse, does not show fatty liver phenotype throughout its lifespan [101]. This emphasizes that reaching a certain threshold of SMN can reverse the fatty liver phenotype.

While very little work has focused on brown adipose tissue (BAT) and white adipose tissue (WAT), molecular changes are present in these tissues. As part of a broader study, KLF15 mRNA is misregulated in both BAT and WAT [102]. A few clock genes (controlling circadian rhythm) showed abnormal expression in these two tissues as well [103]. Using a microarray screen, symptomatic BAT tissue in Taiwanese *(Smn^∆7/∆7^;SMN2^+/+^*) mice displayed a wide array of molecular changes that appeared modulated by light therapy [103]. Additionally, *Smn* expression in early life may be dependent on circadian rhythm in BAT, WAT, and liver tissues [103]. On the other hand, WAT appears to have increased levels of lipogenic genes in sedentary Taiwanese *(Smn^∆7/∆7^;SMN2^+/+^*) mice, which was reversed by exercise [56]. Adipokine levels remained unchanged in another study [30].

#### 5.1.1. Comparison to Models of Denervation and Other Diseases

##### 5.1.1.1. Spinal and Bulbar Muscular Atrophy

Spinal and bulbar muscular atrophy (SBMA) is a motor neuron disease affecting individuals in their adulthood [104]. It is caused by a trinucleotide repeat in the androgen receptor gene located on the X chromosome. As its name indicates, it generally presents with progressive weakness of bulbar and skeletal muscles of individuals in their 40s, but the severity is directly related to CAG repeat length [104,105]. Compound muscle action potential (CMAP) and motor unit number estimation (MUNE) are abnormal in many patients, but not all [104,106]. Compared to SMA, the denervation of skeletal muscle and α-motor neuron loss is expected, but the patients are much older [107].

Significant metabolic abnormalities are noted in SBMA. Reports highlight a certain degree of dyslipidemia in SBMA patients, with most commonly high low-density lipoproteins (LDLs), TGs, and total cholesterol (TC) [104,105,106,108,109,110,111]. Some patients in these studies were already on lipid-lowering and anti-diabetic regimens [104,108]. Most recently, a cohort displayed abnormal TC (51%), LDL (38%), TG (38%), and high-density lipoproteins (HDLs) (77%) [112]. In the latter study, when the SBMA cohort is taken as a whole, the mean values obtained for these parameters (TC, HDL, TG, fasting glucose) are not significantly different from an age-matched control cohort [112]. Some of these findings may represent a subpopulation of patients or simply the normal aging process (similar to the normal population). However, imaging identified signs of fatty liver in more than 79% of the patients [109,112]. Fatty liver was not associated with the length of the repeat, but an association was seen with the body mass index of the participants [109].

The mechanism underlying fat metabolism in SBMA is largely unresolved. Very little information is available in preclinical models to our knowledge. Many studies rely on metabolic pathways in the skeletal muscle as it is deemed a primary disease target. Lipidomic analysis revealed enhanced lipid metabolism and impaired glycolysis in muscle prior to denervation [113]. Conditional hepatic knock-out of the androgen receptor leads to steatosis in male, but not female mice fed a high-fat diet (HFD) [114]. However, it took up to 16 weeks with an HFD and 40 weeks on normal chow and was of macrovesicular type [114]. In addition, these mice developed insulin resistance and hyperglycemia. This is unlike SMA mice, where rapid onset (days) microvesicular predominates and hypoglycemia exists [30,31]. A recent review suggests a potential central role for testosterone in the metabolic picture of SBMA [115]. Francini-Pesenti (2020) proposed that androgen receptor malfunction may foster an environment enhancing metabolic syndrome, type 2 diabetes, adipocyte differentiation, and a consequent high burden of fatty acid in the circulation [115]. Such a hypothetical scenario shows similarities to SMA preclinical models (also, see Section 7 on “Metabolic crosstalk and proposed mechanisms”).

Overall, metabolic abnormalities appear more overt in SBMA patients than in SMA patients. Yet, the aging process may bring these abnormalities naturally and cloud the true contribution to SBMA disease pathogenesis. In the SMA population, it is expected that abnormalities in these measures are rare in children, hence removing this confounding factor.

##### 5.1.1.2. Amyotrophic Lateral Sclerosis

Amyotrophic lateral sclerosis (ALS) is a progressive neuromuscular disorder affecting both upper and lower motor neurons [116]. It is a late-onset disease that affects people between the ages of 40 and 70. ALS is broadly categorized into two types: familial (10%) and sporadic (90%). ALS shares several clinical and pathological features with SMA [117,118]. Despite the differing genetic triggers and the older age of onset, ALS and SMA both share the loss of somatic motor neurons and innervation to voluntary skeletal muscles.

A growing body of evidence suggests energy metabolism dysregulation in ALS patients and animal models [119,120]. Systemic metabolism and metabolic stress have been linked to increased vulnerability of motor neurons and correlate to ALS disease course. As in some types of SMA, reports highlight increased energy expenditure [121] and decreased energy uptake, which lead to reduced fat depots and body weight loss in ALS [122,123,124]. Of note, contrasting findings have been reported (reviewed in [125]). Both hyperlipidemia [126,127] and hypolipidemia have been described in ALS patients, the latter being more prominent in males than females [128,129,130]. The hyperlipidemic phenotype has been associated with favorable outcomes [125,126]. Elevation in levels of sphingolipids and cholesterol esters were noted in the spinal cords and skeletal muscle of an ALS mouse model, further indicating possible lipid alterations [131,132,133,134]. Fatty liver appears to be a common trait of ALS as well [126,127], although this is not seen in a commonly used mouse model of ALS [31]. Moreover, a low-fat diet nearly doubled the lifespan of *Smn^2B/−^* mice [41] and may be beneficial for individuals living with SMA. In contrast to these findings, high caloric [120,135] and ketogenic diets [136] delayed disease onset and extended survival in mouse models of ALS, while caloric restriction shortened the lifespan of superoxide dismutase 1 (SOD1) mice [137]. To support these results, it has been shown that diabetes in ALS patients may in fact delay motor symptoms [138].

At the current time, it appears that there is not much overlap between SMA and ALS metabolic intricacies surrounding fat metabolism.

##### 5.1.1.3. Spinal Cord Injury

Spinal cord injury (SCI) is a partial or total loss of neural signal transmission across and below the level of injury. Loss of somatic and autonomic control leads to a reduction in physical activity, deterioration of body composition and metabolic profile [139,140,141,142]. Depending on the severity of the injury, patients with SCI, like SMA patients, will often experience bone loss and muscle atrophy [143,144,145]. Post-SCI, rapid and dramatic loss of muscle mass occurs, estimated at about 15% loss of lower limb lean mass in the first year [145,146].

Comparable to SMA, muscle atrophy after SCI can result in reduced metabolic rate, increased fat storage, and increased risk of metabolic alterations [147]. Reports of fat mass changes are mixed [148,149]. These inconsistencies are due to patient-specific variables and inaccuracy in assessment methods [148,149].

Lipid profiling in SCI patients also showed that the most common abnormality is decreased levels of serum HDL cholesterol [150,151]. A high percentage of SCI patients has at least one measure of dyslipidemia [151]. While other measures of dyslipidemia (LDL, TG, TC) are observed in SCI, they do not seem to be more prevalent than in the normal population in some studies [151,152]. Liver steatosis is reported in about 20–50% of patients suffering SCI [153,154]. It is thought that many factors can contribute to this, including alcohol use [153] as well as androgen deficiency [154]. Increased levels of circulating non-esterified fatty acids increase their influx into hepatocytes and contribute to the establishment of insulin resistance in the liver [155]. A similar phenomenon is observed in mouse models of SMA [30].

##### 5.1.1.4. Spinal Muscular Atrophy with Progressive Myoclonic Epilepsy

Spinal muscular atrophy with progressive myoclonic epilepsy (SMA-PME) is a rare form of SMA, characterized by myoclonic and generalized seizures with progressive neurological deterioration [156,157,158]. SMA-PME, like SMA, is marked by progressive degeneration of spinal motor neurons, muscle atrophy, and paralysis [159,160,161,162]. It is caused by a mutation in the N-acylsphingosine amidohydrolase 1 (ASAH1) gene, which encodes a ceramidase [159]. Ceramide is synthesized in the endoplasmic reticulum and participates in various cellular events as a lipid mediator. Ceramidase, the dysfunctional enzyme in SMA-PME, is responsible for the catabolism of ceramide and plays a role in sphingolipid metabolism [158,163]. The fatty acid residual from ceramide breakdown is then used to produce myelin. The absence of proper myelin formation leads to nerve cell damage [164]. This is an interesting disease to contrast as its pathogenic etiology stems from an enzyme involved in lipid metabolism and leads to a phenotype that resembles SMA. However, to our knowledge, studies investigating abnormal fatty acid metabolism outside of the nervous system are lacking in SMA-PME, making any comparison to SMA difficult.

### 5.2. Glucose Metabolism

Concern over glucose metabolism in SMA first surfaced in a case series presenting recurrent hospital admissions for hypoglycemia in two patients [165]. Glucose levels in these individuals halved in 12 h without changes in regulatory hormones, but increased ketones were present. This was not seen in muscle disorders such as Duchenne muscular dystrophy, despite a 30 h fast [166]. However, it appears that patients with muscle wasting (muscle mass around 10% of body mass) are more prone to low blood sugar [167]. As such, SMA patients and families were strongly recommended to optimize the feeding schedule to reduce fasting time [50].

In preclinical studies, the *Smn∆7* mice showed hypoglycemia and increased β-hydroxybutyrate (BHB) on two different diets [168]. The *Smn^2B/−^* mice showed hypoglycemia at late stages despite sustained glucose levels upon fasting. Administration of an intraperitoneal glucose tolerance test (IPGTT) suggested that *Smn^2B/−^* mice develop glucose intolerance over time [40]. This was perhaps caused by the predominance of glucagon-producing α-cells in the pancreas at the expense of insulin-producing β-cells [40]. Consequently, the *Smn^2B/−^* mice displayed hyperglucagonemia but normal insulin levels. While blood sugar values from patients were difficult to interpret in this study, autopsies revealed a similar pancreatic islet phenotype [40]. These changes in the pancreas are thought to contribute to the development of NAFLD in *Smn^2B/−^* mice, as hyperglucagonemia and increased CREB (cAMP response element-binding protein) phosphorylation could be driving a surge of energetic substrates in the bloodstream [31]. Sustained low blood sugar may drive hyperglucagonemia [30,31]. Alternatively, the intrinsic pancreatic defects of the α-cells could be the cause. *Smn^2B/−^* mice also show hepatic insulin resistance [30]. In the Taiwanese *(Smn^∆7/∆7^;SMN2^+/+^*) mice, plasma glucose levels were decreased as early as P4 [83], and glucose intolerance was also observed after glucose loading, which was improved by exercise [56]. In the longer-lived model, no hypoglycemia, changes in insulin sensitivity or levels were noted [56]; rather, fasting hyperglycemia was observed during the aging process. Unfortunately, glucagon levels were not measured.

Interestingly, heterozygous animals for SMA (carrier), otherwise asymptomatic from a neurological standpoint, are susceptible to metabolic defects upon dietary challenge (HFD) and aging [169]. However, the metabolic changes seen under these conditions showed some differences with young SMA mice. While similar increases in the phosphorylation of protein kinase B (PKB also known as Akt) were present, the *Smn^+/−^* animals did not display hyperglucagonemia [169]. Instead, they displayed hyperinsulinemia with congruent histological changes in the pancreas [169]. Enhanced phosphorylation of CREB was also shown in different mouse models and settings, potentially highlighting an enhanced pathological sensitivity to glucagon as a common feature [169]. A small cohort of patients did not display abnormal glucagon levels [170]. However, glucagon sensitivity would be challenging to prove in human subjects. As such, it remains possible that CREB activity could be perturbed independently of glucagon levels. CREB activity could also be unique to cell type, as it is low in SMN-depleted neural stem cell (NSC)-34 cells, a mouse motor neuron-like cell line [171]. At this time, it is unclear whether this may predispose SMA carriers (heterozygous individuals) to metabolic abnormalities. To the best of our knowledge, the SMN gene has not surfaced in association studies with diabetes [172,173,174,175,176]. However, it has been associated with a potential role in one of the consequences of diabetes, namely, diabetic polyneuropathy [177].

A state of altered blood glucose may have metabolic consequences for SMA patients, whether treated or untreated. There are a few case reports showing ketoacidosis in the presence but also in the absence of concurrent diabetes [170,178,179,180,181,182,183]. Multiple reports of untreated SMA patients with blood sugar abnormalities have surfaced. A small pilot study showed that out of six children with SMA type II, three had impaired glucose tolerance, and five were insulin-resistant [170], the latter observed in 53% of patients in a more recent study [184]. Similar findings were observed in the glucose tolerance test in seven out of 15 SMA type II and III patients, while 29.7% of their cohort met laboratory criteria for prediabetes [93]. Glucose levels seem to fluctuate greatly in SMA. One patient had hypoglycemia during the oral glucose tolerance test (OGTT) [170], while another study identified no change in fasting glucose [184]. Out of 12 SMA type I patients who unfortunately passed, 10 had at least one glucose value above normal upon chart review [32]. In contrast, a study of 45 SMA type I patients showed that hypoglycemia occurred in 17 of them after fasting 4–6 h, whether for a planned procedure or during an acute illness [185]. This appeared to be independent of their disease severity or respiratory status [185].

Median glycated hemoglobin subunit alpha 1 (HbA1C) of SMA patients in two studies was around 4.9 [31,93]. Another study measured HbA1C in 15 patients, but no objective values were provided apart from being mostly qualitatively normal [184]. It is important to note that HbA1C represents blood glucose over the prior three months and may not adequately highlight the difference in glucose handling in acute settings. Moreover, HbA1C is generally widely used for diabetes diagnosis and monitoring, where important changes noted involve higher ranges of HbA1C. On the other hand, low HbA1C values have been associated with all-cause mortality ([186] HbA1C < 4%, [187] HbA1C < 5%, [188] HbA1C < 5%) in various settings ([189] HbA1C < 5.1%, [190] HbA1C < 5.0%), liver disease ([191] HbA1C < 4%), cancer ([192] HbA1C < 5.0%) and disability in non-diabetic individuals [193]. However, this concept has been debated [194] and most of these studies have been performed in an adult population. Hence, it is unclear whether this would apply to the SMA population. Moreover, causes for low HbA1C may vary and can include anemia and related causes and liver disease, among others [195], which could be a confounding factor.

The possibility of glucose abnormalities in SMA patients should remain a consideration in a newly treated population where most patients may live well into adulthood. At the moment, the issues with glucose metabolism, if any, remain largely undescribed. This stems from differences in the abnormalities observed in different model systems and cohorts of patients. It perhaps is not unexpected as the SMA spectrum is exceptionally varied and these abnormalities may only apply to a subset of patients.

#### 5.2.1. Comparison to Models of Denervation and Other Diseases

##### Spinal and Bulbar Muscular Atrophy

Glucose metabolism abnormalities are also frequently reported in SBMA patients. In a studied cohort of patients with SBMA, 49% had increased fasting glucose levels and 66% had metabolic syndrome, based on a homeostatic model assessment insulin resistance (HOMA-IR) [112]. Several other studies supported these findings with elevations in fasting glucose, increased HbA1C above reference limits, and evidence of insulin resistance [108,109,111]. However, Nakatsuji et al. (2017) found that the mean fasting glucose and HbA1C values were not different from the control population despite some evidence of insulin resistance [196]. They also reported lower protein levels of insulin receptors in autopsies of skeletal muscle from patients with SBMA, compared to controls [196]. Similarly, the prevalence of diabetes in an SBMA cohort (11–18%) does not seem to be higher than in the general population of the same age as recorded by the Centers for Disease Control and Prevention (14.8–17.5%) [111,197,198].

The exact mechanism behind the metabolic alterations observed in patients with SBMA remains to be clarified. However, androgen insensitivity causing insulin resistance in both liver and muscle is the leading proposed mechanism. While insulin resistance is a common aspect of SMA and SBMA, SBMA baseline glucose homeostasis appears to be different from SMA. Hyperglycemia is relatively rare in preclinical SMA models, while median HbA1C is relatively low in SMA patients [31,93], in addition to their being at risk for hypoglycemia [185]. However, higher glucose levels may be observed in select patients [93]. Perhaps the longer-lived treated SMA patients may reveal differences as they age.

##### Amyotrophic Lateral Sclerosis

Defects in glucose metabolism have also been observed in the cortex and spinal cords of SOD1 mouse model of ALS [199,200,201]. Moreover, reduced muscle glucose uptake and glucose intolerance have also been observed in the SOD1 mice [202,203]. However, spinal cords from end-stage mutant SOD1 mice and ALS patients have elevated concentrations of glycogen, suggesting decreased ability to metabolize carbohydrates [132]. Proteomic studies have also revealed downregulation in glycolysis and malate–aspartate shuttle components in ALS skin fibroblasts [204] along with increased expression of respiratory chain proteins in ALS motor neurons [205].

In a cohort of ALS patients, seven out of 18 patients had abnormal oral glucose tolerance tests and significantly higher plasma glucose post-glucose tolerance tests compared to controls [206]. Another study identified that 33% of patients with impaired glucose tolerance also had elevated free fatty acids levels compared to patients with normal glucose tolerance [202]. Plasma glucagon concentrations were 240% higher in SOD1 mice compared to control without any changes in insulin concentration levels [207]. Along with this finding, 22% of pancreatic β-cells were reduced, while no difference was noted in α-cell reactivity. Glycogen levels were elevated in the livers of SOD1 mice, despite the increase in circulating glucagon, suggesting impaired glucagon signaling in the liver [207]. Interestingly, gene profiling revealed that the liver might have been utilizing fatty acids as an energy source instead of glucose.

Despite the reports that impaired glucose tolerance and diabetes mellitus may be more frequent in ALS patients [202,208,209], pre-morbid type II diabetes was associated with a four-year delayed onset of ALS or a decreased risk of ALS altogether [210,211], although this has been debated [209,212]. The pathways and mechanisms linking glucose metabolism to disease pathophysiology remain to be elucidated. Notably, ALS patients do not have hyperglucagonemia [213]. ALS and SMA patients share many clinical and molecular features, including decreased glucose tolerance and insulin resistance. Defects in glucagon and insulin-producing pancreatic islets, as well as increased circulating levels of glucagon, albeit not as profound as in SMA, are significant observations made in both motor neuron diseases.

##### Spinal Cord Injury

Carbohydrate metabolism is altered, and glucose intolerance occurs more frequently in patients with SCI [150,214,215] (and reviewed in [216]). The level of intramuscular fat (IMF) is a good predictor of plasma glucose during oral glucose tolerance tests [217]. Significantly higher serum glucose concentration and diabetes mellitus were also reported in subjects with chronic SCI [214,218]. Following an oral glucose challenge, half of the patients in this cohort had hyperinsulinemia, suggesting insulin resistance [218], and significantly higher glucose uptake than non-SCI controls [219]. The impairment of glucose homeostasis among SCI patients differs from observations in human and mouse models of SMA.

### 5.3. Amino Acid Metabolism

Little is known about amino acid metabolism in SMA. Few studies have reported decreased levels of specific amino acids (alanine, phenylalanine, and branched-chain amino acids) with hypoglycemia upon fasting in SMA type II patients [165,170]. An interesting hypothetical link between hypoglycemia and exhaustion of the amino acid used for gluconeogenesis from the reduced muscle mass in SMA was put forward to explain fasting hypoglycemia [165]. However, objective findings supporting this hypothesis are currently lacking. Branched-chain amino acids (BCAAs) are drastically decreased in the serum of a symptomatic SMA mouse model [102]. Dietary supplementation of BCAAs improved weight gain and survival in the Taiwanese *(Smn^∆7/∆7^;SMN2^+/+^*) mice [102]. Interestingly, some families caring for children with SMA have turned to a diet known as “The Amino Acid diet” (https://www.aadietinfo.com/ (accessed on 31 May 2021)), which was initiated by an SMA caregiver in the hope of improving SMA symptoms. According to their website, it is thought to subjectively improve respiratory health, constipation, strength, and function. These observations remain subjective due to a lack of studies on amino acid metabolism in SMA nutrition. Moreover, this diet can lead to health consequences [220] if not carefully monitored. Given the very little information on this topic in the SMA field, we did not pursue comparison with other neuromuscular diseases.

## 6. Hormonal Regulation

A description of hormonal regulation that may affect metabolism in SMA has been largely lacking. One study recently identified a high prevalence (15/35 patients) of hyperleptinemia in SMA patients, most being underweight [184]. Interestingly, hyperleptinemia was associated with lower motor functions [184]. In contrast, leptin levels were unchanged in a preclinical model of SMA [30]. A recent cross-sectional study revealed precocious pubarche in 24% of an SMA patient cohort without any changes in levels of androgens [221]. This was associated with a marked decrease in muscle mass and a significant increase in insulin resistance in SMA type I and type II patients [221].

In preclinical models, IGF1 levels were invariably low [11,30,222,223]. IGF1 binding proteins were also misregulated, more particularly IGFals but not IGFbp3 [11,30,223], the latter showing contradicting results in one study [222]. Low IGF1 may contribute to the overall small size of SMA mice [11]. Differential expression of IGF1-R is seen in muscle biopsies of SMA type I and type III patients. A study of 15 SMA patients showed lower IGF1 than the control group [224]. However, two other studies found IGF1 to be normal in SMA patients when compared to standard values [170,225]. Interestingly, early studies overexpressing IGF1 in pre-clinical models in various contexts only led to modest improvements in motor phenotype, while metabolic readouts were not assessed [226,227,228].

### Comparison to Models of Denervation and Other Diseases

Alteration in IGF1 levels is a feature of other motor neuron-related disorders. In SBMA, there is a reduction in the plasma levels of IGF1 synthesized by the liver [109]. Despite the different genetic causes of SMA and SBMA, both exhibit liver damage and functional deficit in the IGF1 pathway, suggesting a potential link between fatty liver, insulin resistance, IGF1, and neuromuscular damage.

ALS patients have reduced serum levels of IGF1 as well [229]. They are suggested to correlate with disease progression and degeneration of motor neurons [230]. Interestingly, ALS patients with higher levels of IGF1 had prolonged survival [231,232]. The same trend is observed in leptin levels [233]. Leptin levels in ALS patients correlate with fat mass, increased fatty acid oxidation, and lipolysis [234,235]. This may explain why a high-fat diet may indirectly offer protection by increased energy stores in ALS patients.

Similarly, SCI leads to reduced levels of IGF1 [215,236,237], thought to be secondary to increased FM accumulation in both human and animal models [236,238]. Despite FM accumulation post-SCI, leptin was downregulated in a mouse model of SCI, and acute administration of leptin improves the expression of neuroprotective genes and motor function [239].

## 7. Metabolic Crosstalk and Proposed Mechanisms

The study of metabolism is intricate. Its complexity is multiple-fold. All cell types perform metabolic activity to some extent, yet their contribution or molecular machinery may differ. The organs with a major metabolic role include the GI tract, liver, pancreas, skeletal muscle, and white and brown adipose tissues. These tissues are distant from each other and communicate through various signaling factors, affecting each other’s functions. Metabolic function can be further modulated by an array of hormones that are quickly induced or repressed and may show circadian rhythmicity. In the context of SMA, the idea that each cell type involved may be uniquely affected by SMN depletion adds to this complexity. As such, identifying a potential mechanism underlying a metabolic phenotype requires a holistic approach that incorporates all of these variables. The initiating event(s) leading to the SMA metabolic phenotype as we know it are not easily identifiable. Using the development of the NAFLD phenotype in preclinical models of SMA as a starting point, many findings can be unified. NAFLD can be described as a deficit between input and output of fatty acids (see Figure 2), and some mechanisms such as insulin resistance can exacerbate this.

### 7.1. Fatty Acid Overloading

It is important to acknowledge that the SMN-depleted liver is likely susceptible to fatty acid overload. Recently, an RNA-seq study highlighted various splicing changes secondary to SMN depletion in the liver [100]. The liver is thought to have the most widespread transcriptomic changes compared to brain, spinal cord, and skeletal muscle tissues [100]. Likewise, proteomic changes are most prominent in the liver, as early as in utero [240]. Nevertheless, the link between current hepatic phenotypes observed and these transcriptomic and proteomic changes has not been uncovered. Intron retention appears to be a common occurrence in the setting of SMN depletion. U12-dependent intron retention common across all tissues examined yielded a set of seven genes, of which none appears to have a major role in metabolism [100]. Other changes, perhaps specific to the liver, could contribute to the phenotype. Indeed, some of the pathways significantly downregulated by GO pathway analysis (lipid metabolic process, cellular lipid metabolic process, among others) and Kyoto Encyclopedia of Genes and Genomes (KEGG) pathway analysis (fatty acid metabolism, fatty acid degradation, type I diabetes mellitus, among others) of the transcriptomic results by Doktor et al. (2017) point toward a host of potential culprits that could be the target of future studies [100]. Proteomic analysis as soon as at birth identified multiple pathways involved in metabolism and mitochondrial function [30], while hepatic in utero proteomic analysis did not identify clear metabolic pathways [240].

In addition to the possible intrinsic liver changes described above, indirect stimuli from other metabolic organs could contribute to pathogenesis. In *Smn^2B/−^* mice, we believe that the input of fatty acids to the liver is directly related to pancreatic and WAT contribution. Under normal conditions, glucagon acts in the fasting state to allow continuous energy substrate availability by glycogen breakdown, gluconeogenesis, and lipolysis of peripheral WAT. The extreme hyperglucagonemia and consequent molecular activation of CREB are likely contributory to the excess non-esterified fatty acids in the circulation from possible WAT lipolysis [30]. Interestingly, SMA patients display a significant increase in free fatty acids upon fasting [170], although it is unclear whether it is increased at baseline. This mechanism is generally contributory to 60% of the lipid accumulation in the context of NAFLD [99]. A similar mechanism has been proposed in SBMA, where androgen receptor malfunction leads to increased fat mass and secondary increase in free fatty acid production in the circulation [115] (see Section 5.1.1.1). Dietary intake is unlikely to be involved as diet modulation had no effect on hepatic lipid accumulation [41] but did modulate other measures such as ketones. De novo lipogenesis in SMA was not formally studied and should not be discounted.

At this time, the reason behind the nearly 40-fold increase in glucagon is primarily speculative. The idiopathic expansion of glucagon-producing α-cells in the pancreas of *Smn^2B/−^* mice may pathologically express excess glucagon [40]. The reason for this cellular change has yet to be resolved. The persistent hypoglycemia seen in *Smn^2B/−^* mice [30,31] could induce glucagon expression. Another possible reason for the increase in glucagon may be the difficulty in reaching appropriate nutrition due to the SMA motor phenotype. While fasting/starvation cannot be overlooked due to limitations in the ambulation of SMA patients, glucagon induction seems to be present before motor neuron loss and motor impairment in *Smn^2B/−^* mice [241]. Leptin, a general indicator of energy reserve, can modulate food intake [242]. As such, its secretion can help identify overfeeding and fasting states. Leptin was unchanged in *Smn^2B/−^* mice [30]. Nevertheless, future studies should compare findings in SMA mice with a severe starvation model to dissect the contribution linked to SMA vs. starvation. Of course, a combination of these factors likely explains the hyperglucagonemia. Numerous other factors can modulate glucagon expression, including amino acids, gut peptides, and neuronal signals [243]. GLP1 is an inhibitor of glucagon, yet seemingly has limited effect in *Smn^2B/−^* mice, where it is up-regulated similarly to glucagon [30]. However, glucagon expression can also be due to α-cells in the setting of hyperplasia [244]. It is interesting to note that while amino acids can induce glucagon expression, hyperglucagonemia can itself lead to low levels of amino acids [243,245]. Interestingly, low amino acid levels are seen in SMA [102] and in SMA patients during fasting [170]. To cloud the picture further, NAFLD and related phenotypes are associated with hyperglucagonemia [243], raising the question of which one comes first. To note, a small cohort of six SMA patients did not have hyperglucagonemia during OGTT [170]. More research is required to better understand hyperglucagonemia in SMA.

It is unclear whether the lipolysis is only driven by glucagon or accelerated by WAT dysfunction and/or inflammation. No change in circulating adipokines has been observed in *Smn^2B/−^* mice, including leptin. However, a subset of SMA patients have hyperleptinemia, which is proposed to contribute to enhanced lipolysis and subsequently may lead to liver steatosis [184]. A recent study identified abnormalities associated with the circadian rhythm of important genes of WAT and BAT tissue physiology [103]. Their contribution to the NAFLD phenotype is currently unknown. In milder SMA (type II–III) patients, a recent study proposed that the establishment of liver steatosis may be pretty similar to NAFLD based on the fact that the patients identified to have fatty liver in their cohort were also obese and had prediabetes and dyslipidemia [93].

The denervated muscle in SMA likely enhances the increased fatty acid burden in the circulation. This is mainly due to its non-functional state and consequently diminished uptake of energy sources for maintenance. In addition, the atrophic muscles have intrinsic metabolic program changes. They have reduced capacity for fatty acid oxidation, with a shift toward glucose used as a primary source of energy [246,247].

### 7.2. Fatty Acid Clearance

Fatty acid clearance can be analyzed at different levels. In the circulation, enhanced levels of energy substrates (including fatty acids) are likely due to the reduced utilization of the denervated and eventually wasted SMA muscles (as described above). This leads to a relative increase in substrate availability in the circulation, which may eventually be taken up by the liver. Additionally, it is unclear whether insulin sensitivity is impaired in skeletal muscles (or adipose tissue), which would greatly change their mode of metabolism as well. Hepatic insulin resistance is evident and likely contributes to the current pathogenic NAFLD progression. Future studies should focus on identifying whether insulin resistance is systemic and identify its consequences. At the local level, hepatic lipid clearance through mitochondrial β-oxidation was evaluated and noted to be hyperactivated, likely as compensatory to lipid overload. Nevertheless, this assay was performed using an equal number of mitochondria [30]. It is possible that, as a whole, β-oxidation is reduced as mitochondrial content seems reduced in the *Smn^2B/−^* mouse model. We observed evidence of increased microsomal oxidation, an atypical type of oxidation that is generally not used by the cell unless β-oxidation is affected. This is particularly interesting as one of the by-products of this method of oxidation, dicarboxylic acids, is observed in SMA patients. The inability to secrete lipoprotein may also foster increased lipid storage. Interestingly, endocytosis was recently identified as a defective pathway in SMN-depleted cells [248]. The data for defects in exocytosis are sparser. Indeed, the inability to take in lipoprotein in the peripheral tissues would also worsen the circulatory density of lipids. This area of research currently remains untapped.

In a similar fashion, exercise is generally seen as a protective entity in metabolic disorder, but also in neuromuscular disease, for various reasons including metabolic rate and substrate utilization [249]. In general, most reports in the SMA field focus on the effect of exercise on motor function (a topic outside of the scope of this review) and there is not much information on its influence on the metabolic dysfunction in SMA. One study reported improvement in the energy metabolism of mice with mild SMA after low-intensity running and high-intensity swimming [56]. More specifically, the exercise regimen improved mitochondrial enzymatic activity in the fast-twitch muscles, adipocytic deposition, glucose homeostasis and energy production [56]. Recently, it was shown that exercise could modulate SMN2 exon 7, but there was no change in SMN protein expression in skeletal muscles [250]. Overall, the specific effect of exercise on metabolic outcomes and their mechanism, whether SMN-dependent or independent, remains unresolved. More work needs to be done in this area to assess the overall benefit of exercise.

Overall, it is important to note that the mechanism put forward above relies on data from the *Smn^2B/−^* mouse model (on which most research was based) and isolated clinical studies. Future research should aim at confirming or disproving these mechanisms in other mouse models to further refine our understanding of the metabolic component in SMA. Continued effort toward translational research and identification of the SMA population prone to metabolic issues should be a priority to better medically manage them going forward.

## 8. Nutritional Guidelines for SMA Patients

Overwhelming evidence highlights multilevel metabolic disruption and limitations that come with SMN depletion, which likely leads to inadequacy in reaching appropriate nutrition for SMA patients. This starts from inconsistent quality and quantity of intake, alterations in gut motility, and likely absorption function and continues at the level of macronutrient processing as energy sources. This leads to the metabolic consequences of non-alcoholic fatty liver disease and pancreatic abnormalities in preclinical models, findings that have been thus far reproduced in some but not all SMA patients [30,31,40,93]. It is likely that unlike mouse models, which are generally very homogeneous in their genetics and environment, only a proportion of SMA patients will be susceptible to more prominent metabolic issues. The similarities and differences in metabolic changes between SMA mouse models and findings in human patients are depicted in Figure 3.

Standard of care position statements have continuously requested additional studies to better understand metabolism and nutrition [50,251]. Many studies available rely upon questionnaires. Overall, many possible deficiencies have been described, most synthesized in a recent systematic review [47,252]. There is a lack of prospective studies on the nutritional standards for SMA, and it is unknown what works best. This is a critical area of research as more than one-third of caregivers in a study endorsed modulating the feeding regimen independently, without support from the medical care team [253]. Moreover, a vast number of families use elemental formulas [252,253], on which there is currently no research. While malnutrition is still present in nusinersen-treated cohorts, a combination of adequate nutritional intervention before nusinersen treatment led to better improvement in various measures of nutrition [254]. Of interest, a recent study about the metabolome of SMA patients showed minimal difference upon nusinersen treatment, inferring that centrally administered (i.e., intrathecal) SMA therapeutics may not affect this component of the SMA phenotype [255]. To our knowledge, studies aimed at investigating metabolism upon treatment are not yet available for other treatment options such as Zolgensma or risdiplam. In a phase 3 study (STR1VE), 68% of patients did not require feeding support and 64% maintained weight consistent with age [256]. In-depth study of nutrition has not yet been performed. As such, it is clear that more research is warranted to address these gaps.

Over the years, authors involved in metabolic studies have speculated and suggested possible ideas based on their findings. Most advocate for diets composed mainly of carbohydrates and proteins [39,89,92,165]. This is partly in line with a recent preclinical study demonstrating that a low-fat diet doubles lifespan in the *Smn^2B/−^* mouse model [41]. Low-fat diets with high sucrose content had the best effect in normalizing abnormal readouts (such as ketones), suggesting that this led to lesser reliance on fat as an energy source [41]. However, some discordant studies exist and have been discussed more extensively elsewhere [41,62,257]. An early small study looking at a low-fat/high carbohydrate diet claimed positive results [92], but concerns about its validity remain as this was ahead of genetic testing and the classification of patients was unclear.

High sucrose and low-fat diet could have benefits for SMA patients. Fatty foods do lead to delay in gastric emptying and slower transit time, which may worsen gastroesophageal reflux disease and constipation [258,259]. These two features are already prominent in the SMA population. However, such regimens could also have consequences for the individual. High carbohydrates may allow for the establishment of insulin resistance [260]. Low fat may foster fat-soluble vitamin deficiency [261]. As such, the macronutrient content should be tailored and optimized to the metabolic functions, whether normal or altered in SMA, to ensure the best utilization of energy sources. Despite the initial evidence reviewed here, it remains unclear whether SMN-depleted cells have preferential energy sources based on enzyme level and function. In addition, this may be only applicable to a subset of SMA individuals also carrying a genetic program (polymorphism in other genes), making them prone to metabolic alterations. Investigating various nutritional possibilities for this group of patients is warranted. It may be interesting to see whether this should be applied to all patients or simply those who show metabolic disturbances such as dyslipidemia and fatty liver. Nevertheless, the tests screening for these metabolic alterations are not currently standards of care but should be considered in populations that are longer-lived if a metabolic phenotype is suspected. Long-standing dyslipidemia could have secondary effects on cardiovascular disease, amongst others, as SMA patients age.

## 9. Conclusions

Over the years, it has become clear that SMN depletion leads to a vast array of metabolic issues. The first reports of metabolic disturbance were in human patients, followed by a gap in research on the matter. Most recently, advances in this area were generated in preclinical models of SMA, where most discoveries have been made, with some translational studies corroborating many of the basic research findings in human patients. In this review, we provided the most thorough and up-to-date evidence on this topic. We show that the SMA metabolic phenotype appears mostly distinct, at least from a mechanistic point of view, from other denervating as well as related diseases. This is especially true when considering that SMA is mostly a pediatric disease, in comparison to the comparators used herein, where metabolic alterations are not clouded by the naturally occurring metabolic changes that arise with aging. However, there is a paucity of research attempting to identify the mechanisms underlying metabolic changes in other diseases at this time and more research may help unify some common mechanisms. It is possible that some of the metabolic disturbances are repercussions of the neuromuscular phenotype. Whether unique or secondary to a primary event, it is difficult to determine how these metabolic alterations contribute to the phenotype of SMA patients and affect their long-term outcomes.

Going forward, it will be crucial to decipher some key areas: (1) identifying whether the metabolic defect variability is dependent on the SMN level within the individual mouse models, (2) assessing the mechanism by which metabolic organs interact with each other and affect each other’s phenotype in SMA, (3) identifying the susceptibility factors fostering the development of a metabolic phenotype in SMA patients, knowing that only a subset of SMA patients have such abnormalities, (4) deciphering whether further metabolic abnormalities will arise in the treated population, and finally, (5) how to best manage and limit the added comorbidities and cardiovascular risk in SMA patients throughout their lifetimes.

## Figures and Tables

**Figure 1 ijms-22-05913-f001:**
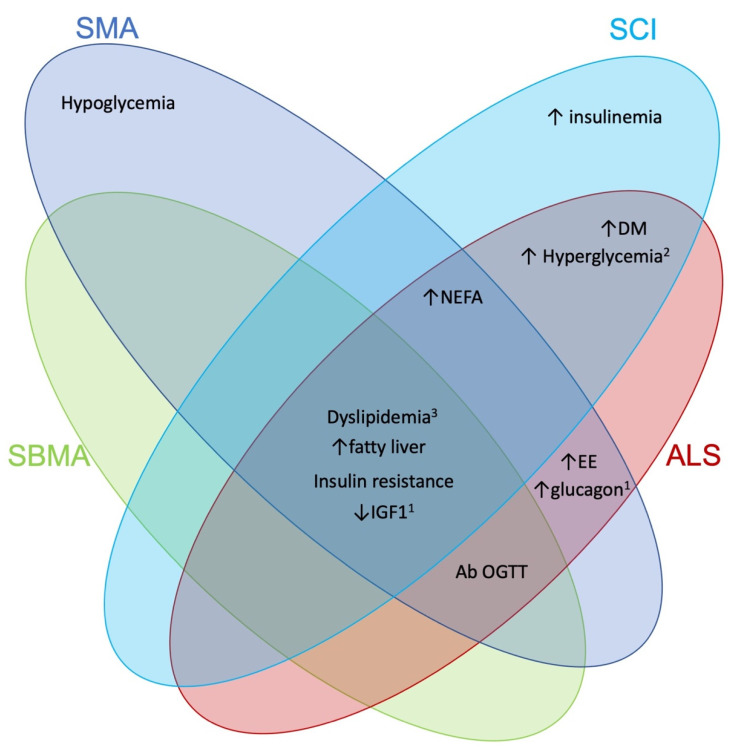
Common metabolic findings within neuromuscular disorders. Venn diagram showing similarities and differences in metabolic changes among different neuromuscular disorders with a denervating/loss of motor nerve input component. Note that studies with mixed results in the current literature will be seen as unchanged within this diagram, given lack of consensus. (1) Some features may only be seen in mouse models of the disease, and (2) in fasting state. (3) Dyslipidemia describes various changes and may not actually be similar between different diseases. For example, it may be hypolipidemia, hyperlipidemia, or differences in frequency or absolute mean of various readouts. DM—Diabetes mellitus, EE—energy expenditure, Ab—abnormal, OGTT—oral glucose tolerance test, HDL—high-density lipoprotein, LDL—low-density lipoprotein, TC—total cholesterol, NEFA—non-esterified fatty acids, black arrow pointing up (↑)—increase, black arrow pointing down (↓)—decrease.

**Figure 2 ijms-22-05913-f002:**
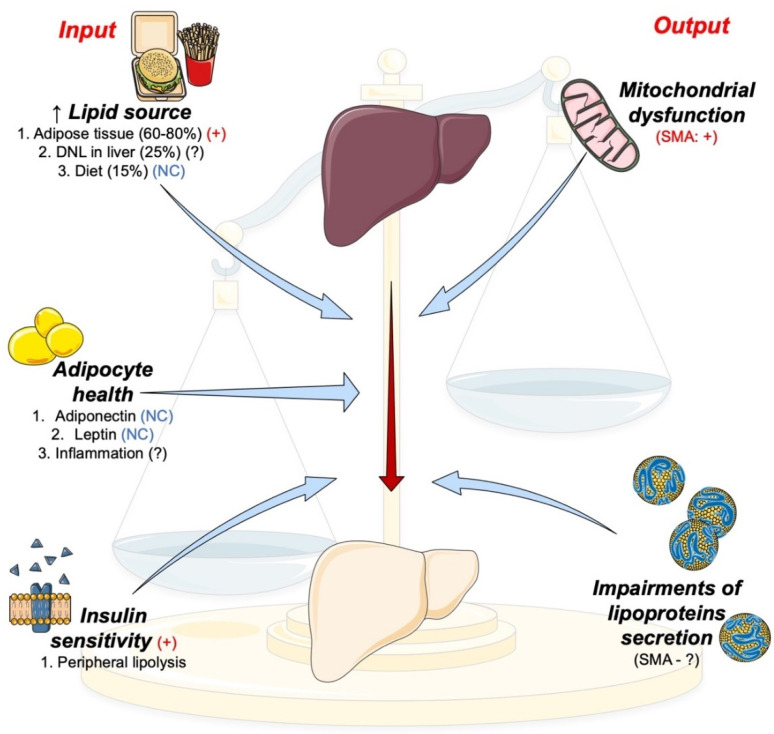
Overview of mechanisms contributing to non-alcoholic fatty liver disease (NAFLD) and their influence in the Spinal Muscular Atrophy (SMA) metabolic phenotype. NAFLD can be seen at the basic level as an imbalance in input and output of fat substrates. The input of fatty substrates can come from various sources, most of which are generally from non-esterified fatty acids from adipose tissues. The input can be influenced by other factors including adipocyte health as well as insulin resistance. On the other hand, the liver deals with these substrates by either using them during fatty acid oxidation or exporting them out of the liver. The contribution of these findings to SMA is described by (+) as found in SMA, (NC) no change seen in SMA, (?) no data in current literature. DNL—de novo lipogenesis. Black arrow pointing up (↑)—increase. The figure was created by the authors using schematic art pieces provided by Servier Medical art, https://smart.servier.com (accessed on 31 May 2021). Servier Medical Art by Servier is licensed under a Creative Commons Attribution 3.0 Unported License.

**Figure 3 ijms-22-05913-f003:**
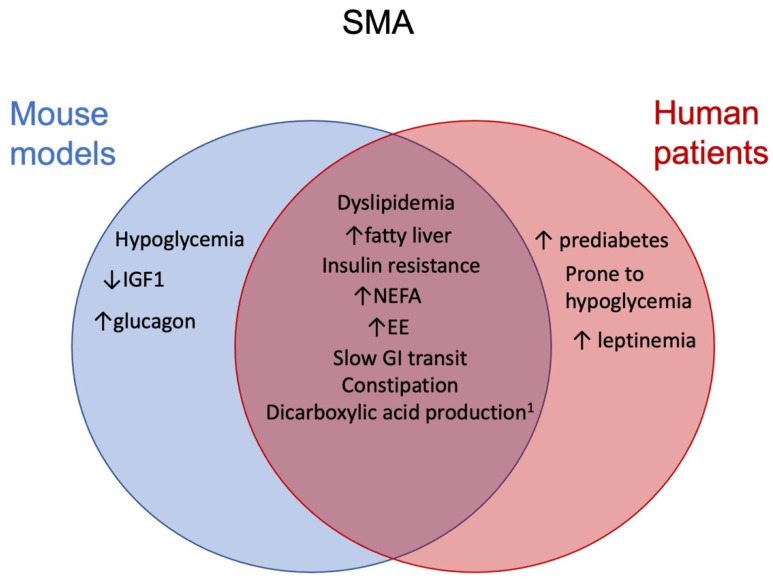
Commonalities between SMA mouse models and human patients. Note that findings in SMA patients may represent a subgroup rather than something found in all patients. (1) Dicarboxylic acid production is inferred by evidence of microsomal oxidation in *Smn^2B/−^* mouse model. EE—energy expenditure; NEFA—non-esterified fatty acids, black arrow pointing up (↑)—increase, black arrow pointing down (↓)—decrease.

## Data Availability

Not applicable

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
