# Peer review of "Metabolic Dysfunction in Spinal Muscular Atrophy"

_ijms, 2021, doi:10.3390/ijms22115913_

Round 1

Reviewer 1 Report

This a well presented and thorough review of the field. There is, as expected a reliance upon data from animal models but this is treated with the care required owing to the possible issues with genetic modification of the mice being not entirely a match with the human situation. It is easy to over interpret the data from animal studies as is shown by the different results from different models. A couple of questions arise from my reading which are not necessarily to be answered in this review. Is there any data on ethnic background and dietary influences? Is there any information on why there is such a high carrier status? In terms of the text itself I think it is important to check the referencing - there are possible instances of incorrect references eg page 7 ref 31. There are examples of juvenile ALS, it is not exclusively adult. 

Author Response

Point 1: Is there any data on ethnic background and dietary influences?

Response 1: Ethnicity-specific information on dietary influences is unavailable. Information on ethnicity has been surveyed in one nutritional studies (e.g. Davis et al., 2013, PMC4334580) however, this study did not comment on any influences or differences according to ethnicity. As such, we have chosen not to discuss this in our review.

Point 2: Is there any information on why there is such a high carrier status?

Response 2: Pericentromeric location and presence of the paralogous segmental duplication likely predispose the region to recombination events, leading to the high observed de novo deletion mutation rate (PMID: 15470363). The carrier frequency also varies among different ethnic groups (pan-ethnic average of 1 in 54) and is suggested to be possibly due to chance variation. Specific details of studies about carrier frequency are outside the scope of this review.

Point 3: In terms of the text itself I think it is important to check the referencing - there are possible instances of incorrect references eg page 7 ref 31.

Response 3: We have verified the referencing. Page 7, Ref 31 (Deguise et al., 2019), is the correct reference. In this paper, the authors show that there is no hepatic fat accumulation in livers of SOD1G93A mutant mice, a model of ALS. This was performed to compare to fat accumulation in livers from Smn2B/- mice to investigate whether denervation (present in both ALS and SMA) was leading to fatty acid accumulation in SMA livers.

Point 4: There are examples of juvenile ALS, it is not exclusively adult. 

Response 4: We have deleted the sentence from the text to avoid confusion.

Reviewer 2 Report

The manuscript “Metabolic dysfunction in Spinal Muscular Atrophy”, by Deguise et al., is a very interesting work giving a wide and comprehensive overview on metabolism affection in SMA. Interestingly, by comparing dysfunctions of metabolic profiles from SMA and other neuromuscular disorders, authors highlighted the uniqueness of SMA metabolism defects as well as common traits with other diseases. The manuscript is original and almost unique in the field and it absolutely deserves to be considered for publication In IJMS. However, before it is published, I have some comments and suggestions for authors for improving, in my opinion, the quality of their work.

The changes I suggest to make are “minimal” but I report them as major comments because I would not want them to be underestimated by the authors. I am sure the authors will be able to resubmit the work in less than 24 hours!    

Major comments

1) Abbreviations, even if known to most (e.g. HbA1C, CDC…) should always be specified the first time they appear in the text.

2) Pag. 3: “Dietary and caloric intake in SMA patients may be lower than what is recommended for healthy peers due to decreased demands [43,62]…….. maintaining bone density and health in this population, especially considering potential intrinsic abnormalities to those pathways [46,47,73]”. Authors should consider and discuss that the low bone density may be also due to the obvious reduced induction skeletal muscle and bone reciprocally act. In fact, while muscle contractions/tractions are necessary for bone developing density and resistance, bone growth and resistance to contraction are crucial for muscle mass and contractile strength development.

3) Pag. 5: “Interestingly, only minor changes were identified in fat classes and chain length in SMA disease-relevant tissues such as the spinal cord and the muscle, revealing that this may be restricted to the liver or the metabolic organs [31]”. Despite this, altered presences of lipid carriers, Apoa1 and ApoE, which may correlate with lipid metabolism, have been recently described in CSF of SMA type 1 patients (Bianchi et al., 2021, DOI: 10.3390/ijms22094329). Nusinersen treatment apparently restored those defective occurrences. Authors are invited to consider this point in the text as it may correlate with lipid metabolism dysfunction in CSF. Impairments in lipoprotein secretions have been even suggested by authors as shown in figure 2.  

3) Authors have repeatedly reported studies on metabolic dysfunctions, both of lipids and carbohydrates, problems of malabsorption and nutrient deficiency, as well as defects of the epithelium and intestinal transit and widespread inflammatory phenomena. The authors could therefore discuss, even simply as future scenarios, the use of probiotics in SMA (marginally mentioned in Martínez Leo and Segura Campos, doi.org/10.1016/j.nut.2019.110609).

Author Response

Point 1:  Abbreviations, even if known to most (e.g. HbA1C, CDC…) should always be specified the first time they appear in the text.

Response 1: Abbreviations have now been defined at their first appearance in the text.

Point 2:  Pag. 3: “Dietary and caloric intake in SMA patients may be lower than what is recommended for healthy peers due to decreased demands [43,62]…….. maintaining bone density and health in this population, especially considering potential intrinsic abnormalities to those pathways [46,47,73]”. Authors should consider and discuss that the low bone density may be also due to the obvious reduced induction skeletal muscle and bone reciprocally act. In fact, while muscle contractions/tractions are necessary for bone developing density and resistance, bone growth and resistance to contraction are crucial for muscle mass and contractile strength development.

Response 2: We’ve now briefly mentioned that reduction in muscle mass and mobility in SMA patients may be influencing bone density.

Point 3:  Pag. 5: “Interestingly, only minor changes were identified in fat classes and chain length in SMA disease-relevant tissues such as the spinal cord and the muscle, revealing that this may be restricted to the liver or the metabolic organs [31]”. Despite this, altered presences of lipid carriers, Apoa1 and ApoE, which may correlate with lipid metabolism, have been recently described in CSF of SMA type 1 patients (Bianchi et al., 2021, DOI: 10.3390/ijms22094329). Nusinersen treatment apparently restored those defective occurrences. Authors are invited to consider this point in the text as it may correlate with lipid metabolism dysfunction in CSF. Impairments in lipoprotein secretions have been even suggested by authors as shown in figure 2.  

Response 3: We agree. We have now mentioned this article in the text (page 5).

Point 4:  Authors have repeatedly reported studies on metabolic dysfunctions, both of lipids and carbohydrates, problems of malabsorption and nutrient deficiency, as well as defects of the epithelium and intestinal transit and widespread inflammatory phenomena. The authors could therefore discuss, even simply as future scenarios, the use of probiotics in SMA (marginally mentioned in Martínez Leo and Segura Campos, doi.org/10.1016/j.nut.2019.110609).

Response 4: We have briefly discussed the emerging role of probiotics in modulating gut microbiota and its potential role in subsequently influencing gut-brain communication and inflammation in SMA in the revised text.

Reviewer 3 Report

Spinal muscular atrophy (SMA) is a neurodegenerative disease affecting children. Previous studies have long focussed on pathologies inside the CNS, but there is increasing but still little evidence that SMA is a multisystemic disorder involving the metabolism of different organs. Nutrition of affected children has long been empirical, but correct use of macronutrients may have an impact on disease progression/severity. In their review, the authors put a focus on this important topic and discuss the current knowledge on metabolic disturbances in SMA patients and mouse models with a main focus on lipid metabolism as well as discuss the need of future (pre-clinical) studies. Furthermore, they compare these findings with those from other related disorders. Based on the current knowledge, the authors finally provide a potential mechanism leading to metabolic changes.

Though the cell type-specific intracellular molecular mechanisms are only little addressed largely due to a lack of studies, I feel that the description/discussion of mechanisms from a systemic perspective (i.e. crosstalk of affected organs) meets the criteria and scope of IJMS and is worth for publication.  

However, I have some minor questions/suggestions:

General:

It is noteworthy that the authors already state the genetic description and severity of the mouse models analysed in parentheses within the text, but non-familiar readers could still be confused. While many models are self-explaining, remaining questions could be: What does 2B- mean? What is the difference between del7 and Taiwanese mice? The authors should briefly explain these models in the text at appropriate position / or supply additional or supplementary box or table / or at least supply a reference for summary.

Specific:

p. 3

I feel that “victims” could be a bit too harsh.

pp. 9, 12, 16:

“Smn+/- animals displayed hyperinsulinemia…” Since insulin/IGF1 and other hormones act upstream of the PI3K/mTOR axis (in line with altered AKT and CREB phosphorylation): Are there any data on altered autophagy-dependent glucose/lipid catabolism in SMA? Could this be a molecular link to the increased microsomal oxidation observed in 2B- mice (p. 16)? Some older studies have tested the effect of IGF1 restoration in SBMA/ALS. Does IGF1 supplementation rescue metabolic/nutritional/fat mass alterations (in case that this has been analysed apart from the muscle phenotypes)?

p. 17:

“Spinraza-treated children still suffer from malnutrition.” Are there already data on the nutritional status of SMA children who received a gene therapy with Zolgensma? Are there pre-clinical data?

Author Response

Point 1: It is noteworthy that the authors already state the genetic description and severity of the mouse models analysed in parentheses within the text, but non-familiar readers could still be confused. While many models are self-explaining, remaining questions could be: What does 2B- mean? What is the difference between del7 and Taiwanese mice? The authors should briefly explain these models in the text at appropriate position / or supply additional or supplementary box or table / or at least supply a reference for summary.

Response 1: We have provided a brief explanation of the key mouse models of SMA on page 2 of the revised text and have provided the relevant references as well.

Point 2: Page 3 - I feel that “victims” could be a bit too harsh.

Response 2: We have changed the wording to soften the language.

Point 3:  Page 9, 12, 16: “Smn+/- animals displayed hyperinsulinemia…” Since insulin/IGF1 and other hormones act upstream of the PI3K/mTOR axis (in line with altered AKT and CREB phosphorylation): Are there any data on altered autophagy-dependent glucose/lipid catabolism in SMA? Could this be a molecular link to the increased microsomal oxidation observed in 2B- mice (p. 16)?

Response 3: To our knowledge, there are no definitive studies on the topic of altered autophagy-dependent glucose/lipid catabolism in SMA.

Point 4:  Some older studies have tested the effect of IGF1 restoration in SBMA/ALS. Does IGF1 supplementation rescue metabolic/nutritional/fat mass alterations (in case that this has been analysed apart from the muscle phenotypes)?

Response 4: Previous studies have shown minimal to modest effect of IGF1 restoration to the SMA phenotype in mouse models. There has not been any study performed to assess if there was any impact on the metabolic aspects of the phenotype. We modified the text accordingly.

Point 5:  Page 17: “Spinraza-treated children still suffer from malnutrition.” Are there already data on the nutritional status of SMA children who received a gene therapy with Zolgensma? Are there pre-clinical data?

Response 5: We have provided the little information available on nutritional status in patients receiving gene therapy with Zolgensma in the revised text. The findings from the STR1VE phase 3 clinical trial showed that 68% of patients did not require feeding support and 64% maintained weight consistent with age. We modified the text accordingly.

Reviewer 4 Report

Deguise at al. present a comprehensive review of metabolic changes in SMA and its animal models. They compare the most important findings to other well-chosen diseases to elucidate a possible pathological overlap and examine their differences. The three figures are aesthetically appealing and clear in their message. Overall, the choice of literature is balanced and robust.

However, as the authors state early in the introduction, SMA has a wide range of onset and severity. The authors choose to follow the 3-tier classification (SMA I, II and III) for most of their review (just once jumping into the “sitters” and “non-sitters”-classification), which is appropriate, but should be explained in the background. In “Baseline anthropometric data […]” the authors briefly hint at how metabolically different the SMA-types might be. Seeing a vast difference in feeding, mobility, energy expenditure and morbidity between the types, the differences between the types should be stated in all chapters of the review. It was also somewhat surprising not to see any commentary on the influence of exercise and mobility on the noted metabolic dysfunction. A distinction between the SMA-types in the respective chapters might help to draw clearer conclusions and make the review more coherent.

Author Response

Point 1: However, as the authors state early in the introduction, SMA has a wide range of onset and severity. The authors choose to follow the 3-tier classification (SMA I, II and III) for most of their review (just once jumping into the “sitters” and “non-sitters”-classification), which is appropriate, but should be explained in the background.

Response 1: Classification of “sitters” vs “non-sitters” has now been defined. 

Point 2: In “Baseline anthropometric data […]” the authors briefly hint at how metabolically different the SMA-types might be. Seeing a vast difference in feeding, mobility, energy expenditure and morbidity between the types, the differences between the types should be stated in all chapters of the review.

Response 2: At the present time, there is not enough data to perform an analysis of metabolic differences within the different SMA types for all other sections in the manuscript. When available, we added a statement concerning this.

Point 3: It was also somewhat surprising not to see any commentary on the influence of exercise and mobility on the noted metabolic dysfunction. A distinction between the SMA-types in the respective chapters might help to draw clearer conclusions and make the review more coherent.

Response 3: While there are a few studies aimed at studying the effect of exercise in SMA, most focus on motor function outcomes (which is currently mixed) rather than  its influence on the metabolic dysfunction in SMA. We added a brief section on the matter in the manuscript.

Round 2

Reviewer 2 Report

The manuscript has been properly modified as required. It is a very interesting work that will make a considerable contribution in the evaluation of metabolic dysfunctions in SMA. I recommend its publication in IJMS.